# Three-Year Intervention Effects on Food and Beverage Intake—Results from the Quasi-Experimental Copenhagen School Child Intervention Study (CoSCIS)

**DOI:** 10.3390/ijerph181910543

**Published:** 2021-10-08

**Authors:** Xuan Ren, Britt Wang Jensen, Sofus Christian Larsen, Jeanett Friis Rohde, Ina Olmer Specht, Birgit Marie Nielsen, Ida Husby, Anna Bugge, Lars Bo Andersen, Ellen Trolle, Berit Lilienthal Heitmann

**Affiliations:** 1Research Unit for Dietary Studies at The Parker Institute, Bispebjerg and Frederiksberg Hospital, 2000 Frederiksberg, Denmark; sofus.larsen@regionh.dk (S.C.L.); jeanett.friis.rohde@regionh.dk (J.F.R.); Ina.Olmer.Specht@regionh.dk (I.O.S.); berit.lilienthal.heitmann@regionh.dk (B.L.H.); 2Center for Clinical Research and Prevention, Bispebjerg and Frederiksberg Hospital, 2000 Frederiksberg, Denmark; Britt.Wang.Jensen@regionh.dk; 3Health Research and Innovation, Centre for Regional Development, Capital Region of Denmark, 2000 Copenhagen, Denmark; birgit.marie.nielsen@gmail.com; 4The Danish Health Authorities, 2300 Copenhagen, Denmark; idahusby.frb@gmail.com; 5Department of Midwifery, Physiotherapy, Occupational Therapy and Psychomotor Therapy, University College Copenhagen, 2200 Copenhagen, Denmark; ABUG@kp.dk; 6Department of Education, Arts and Sport, Western Norway University of Applied Sciences, 6851 Sogndal, Norway; Lars.Bo.Andersen@hvl.no; 7Division of Food Technology, National Food Institute, Technical University of Denmark, 2800 Kongens Lyngby, Denmark; eltr@food.dtu.dk; 8The Boden Institute of Obesity, Nutrition, Exercise & Eating Disorders, Sydney Medical School, University of Sydney, Sydney, NSW 2006, Australia; 9Department of Public Health, Section for General Practice, University of Copenhagen, 1014 Copenhagen, Denmark

**Keywords:** school-based intervention, children, food, beverage, maternal education

## Abstract

The diet of Danish children is often not in accordance with dietary guidelines. We aimed to evaluate changes in the intake of selected foods and beverages during a multi-component school-based physical activity intervention, and to investigate if changes were modified by socioeconomic status (SES). The study included 307 children (intervention group: 184; comparison group: 123) with information on dietary intake pre- and post-intervention as well as on SES. Linear regression models were conducted to assess the effect of the intervention on changes in dietary factors. Children from the intervention group increased their intake of whole-grain bread during the intervention (group means: 6.1 g/d (95% CI: 2.2 to 10.0) vs. 0.3 g/d (95% CI: −3.1 to 3.7) in the comparison group, *p* = 0.04). A significant interaction between SES and group allocation was observed to change in fruit intake (*p* = 0.01). Among children from low SES families, only those from the comparison group decreased their fruit intake (group means: −40.0 g/d (95% CI: −56.0 to −23.9) vs. 9.3 g/d (95% CI: −16.1 to 94) in the intervention group, *p* = 0.006). The present study found no convincing effect of introducing a multi-component intervention on dietary intake except a small beneficial effect on whole-grain bread consumption. However, beneficial intervention effects in fruit intake were found particularly among children from low SES families.

## 1. Introduction

There is a high prevalence of childhood overweightness and obesity in many countries, including Denmark [1,2]. Children with obesity have a high probability of becoming obese adults, and possess a higher risk of developing both short- and long-term adverse physical and mental health [3,4,5,6,7]. A social gradience is observed in many Western countries, with a higher prevalence of overweightness and obesity among children from families with a low socioeconomic status (SES) [8,9,10,11,12]. 

A healthy diet, with regular physical activity, is considered important for the prevention of overweightness and obesity [13], as well as other diseases such as coronary heart disease, type 2 diabetes and some types of cancer [14,15,16,17,18].

The World Health Organization (WHO) and the Danish health authorities recommend increasing the intake of fruit, vegetable, legumes, fish, whole grains and nuts; to limit the consumption of red meat, sugars, salty and total fats; to replace saturated fat with unsaturated fats; and to drink more water [19,20]. However, children and adolescents generally do not comply with these dietary guidelines [14,21]. Most children consume excess amounts of red meat, saturated fat and added sugar, while the intakes of fish, vegetables and whole-grain products are lower than recommended [14,22,23]. In particular, children and adolescents from low SES families may have less healthy dietary habits compared to the recommendations [24,25,26], especially in relation to having low fruit and vegetable intakes and a high intake of sugar-sweetened beverages (SSBs). 

One arena that is considered ideal for introducing health-promoting initiatives to children is within public schools [27], and thus school-based interventions may be efficient as a means of encouraging better diets at the population level [28]. Indeed, previous intervention studies found that both eating habits and attitudes were improved after interventions, for example, related to increased consumption of fruits and vegetables and reduced consumption of SSBs and unhealthy snacks (desserts and starchy foods high in oil) [28,29,30]. However, other studies have failed to show benefits related to fruit and vegetable consumption or have even found decreased consumption despite introducing education programs and improvements in the environment [31,32]. Furthermore, studies have found that dietary habits are related to maternal education level, with more healthy food choices the higher the maternal education [33,34,35]. The Copenhagen School Child Intervention Study (CoSCIS) was started in 2001 and was based on the initiative of a suburban municipality of Copenhagen, Denmark (Ballerup), to improve the diets of and increase physical activity among the local school children. The purpose of the CoSCIS was to evaluate the local initiative. The present study has two aims: (1) to examine the effect of the CoSCIS intervention on the intake of specific foods and beverages and (2) to examine if a potential intervention effect was modified by maternal educational level.

We hypothesized that: (1) the intervention would result in children from the intervention schools having a higher intake of fruit and vegetables and a lower intake of sweets and SSBs than the children from the comparison schools, and (2) that maternal education level would modify the potential effects of the intervention.

## 2. Materials and Methods

### 2.1. Intervention 

The intervention introduced in the local schools in the municipality of Ballerup was multi-component and has been reported in detail elsewhere [24]. Briefly described, it consisted of (1) two additional physical education (PE) lessons (each consisting of 45 mins) per week, (2) additional education of PE teachers, (3) improvement in the physical schoolyard environment, (4) introduction of healthy school canteens, (5) parental involvement and (6) health education in the curriculum. 

The intervention was initiated and implemented by the municipality of Ballerup, while the research team performed pre- and post-evaluations of dietary intake, physical activity level and anthropometrics. The municipality of Tårnby was chosen as a comparison municipality since the sociodemographic characteristics resembled those of Ballerup [36].

Participants, teachers, school leaders and researchers were blinded to the study condition assignment. Prior to the intervention (the children were 6 years old), information on dietary intake and socioeconomic status was obtained from the children and their parents. Weight and height were measured at enrollment by trained researchers. Height was measured to the nearest 1 mm by a Harpenden stadiometer. Weight was measured to the nearest 0.1 kg by an electronic scale (Seca 882, Medical Scales, NY, USA). Habitual physical activity was measured on 4 consecutive days (2 weekend days + 2 weekdays) by the MTI 7164 activity monitor (Manufactory Technology Inc., Fort Walton Beach, FL, USA). Children were required to wear accelerometers at all times, except when sleeping and/or engaged in water activities. Only children with a cumulative activity time of more than 8 hours per day for at least 3 days were included in the analysis. Similarly, this information was collected after the three-year intervention (when the children were 9 years old).

### 2.2. Study Setting and Population

All children entering pre-school in public schools in the municipality of Ballerup (ten schools) served as the non-randomized intervention group while the children entering public schools (eight) in the municipality of Tårnby served as the comparison group (total *n* = 1024 children). The pre-intervention characteristics of all the participants have been reported elsewhere [37,38,39].

Of the invited 1024 children, parents or caregivers provided written consent for 701 (69%) [37,38], 411 in the intervention group and 290 in the comparison group. Of those, 499 children (intervention group: *n* = 292; comparison group: *n* = 207) provided dietary information pre-intervention. Post-intervention, a total of 307 children (intervention group: *n* = 184; comparison group: *n* = 123) provided information on dietary intake as well as on maternal education and were included in the final analyses (Appendix A) [40]. 

### 2.3. Ethics

The study was approved by the Ethical Committee of Copenhagen County (case no. KA00011gm) and all procedures were in accordance with the Helsinki Declaration of 1975, as revised in 1983. Written information about the study was given to all school leaders, teachers and parents prior to the study. Informed consent was obtained from the parents/caregivers of the participants. 

### 2.4. Dietary Assessment 

Parents or caregivers recorded the dietary intake of the children through 7-day food records (developed by the National Food Institute, Technical University of Denmark) both pre- (spring 2002) and post-intervention (spring 2005). The food records were divided into 4 parts: breakfast, lunch, dinner and snacks. Each meal was further divided into beverages, bread, cereals, vegetables, etc. Household measurements and photo series illustrating the serving sizes and 4 or 6 different quantities of common foods were used to assess the amount of food consumed [24,36,37]. 

After completion the food records were scanned using Eyes & Hands version 5.2 and the dietary intake of each child was calculated using the General Intake Estimation System (version 1.000, released 26 February 2010, developed by the National Food Institute, Technical University of Denmark) and the Danish Food Consumption Databank (version 6) [24].

In the present study, we focused on the intake of fruits (fresh, canned, dried fruit and jam, excluding fruit juice), vegetables (fresh, canned and frozen vegetables (including ketchup and fried onion, excluding potatoes)), fruit juice (100% apple juice and 100% orange juice), SSBs (soft drinks (carbonated sugar-sweetened soft drinks) and squash), sweets (chocolate, candy, jelly, caramel and marzipan), white bread (bread reported as white bread in the food records), whole-grain bread (bread reported as whole-grain bread in the food records, including bread with variable content of whole meal and whole grain, and not necessarily in accordance with the official definition of whole-grain bread), dark rye bread (bread recorded as rye bread in the food records, typical Danish bread high in dietary fiber) and fish (fatty fish, lean fish and shellfish) [41]. The food and beverage groups included in the present study were based on the Danish food-based dietary guidelines [42,43].

### 2.5. Socioeconomic Status

Prior to the intervention, the mothers provided information on their education level in a questionnaire using two questions [24]. The first question was “what school education have you achieved or are you currently completing?”. The second question was “do you have a vocational education?”. Based on this information, maternal education level was afterwards grouped into three categories: short education, completed elementary school only (≤10 years); medium education, completed high school (12 years) or a vocational education (3 years); and long education, completed college or university. 

### 2.6. Statistical Analysis

Pre-intervention characteristics of the children participating in the study were assessed according to the intervention status or SES, and differences between the two groups were tested using a Wilcoxon rank-sum test for continuous variables and Chi-squared test for categorical variables.

Linear regression models were conducted to assess the effect of the intervention on changes in dietary factors. First, crude models were conducted with information on outcome, intervention status and baseline/pre-intervention measure of outcome, only. Secondly, adjusted analyses with added information on sex, age and maternal education were carried out. 

Potential effect modification by maternal education status was explored by adding product terms to the models, and subgroup analyses were performed when significant interactions were observed.

Normality of continuous variables and model assumptions (investigating linearity of effects on outcomes, consistency with a normal distribution, and variance homogeneity) were assessed through visual inspection of histograms and residual plots. As some discrepancies from the model assumptions were observed, all results were presented with 95% bootstrap confidence intervals (CIs).

All statistical tests were two-tailed, with a significance level at 0.05. Analyses were performed using Stata SE 14 (StataCorp LP, College Station, TX, USA).

## 3. Results

The analytic study population consisted of 307 children (intervention group: *n* = 184; comparison group: *n* = 123) with complete information on dietary intake and maternal education level. The intervention and comparison groups differed slightly pre-intervention with respect to age and maternal education level (*p* = 0.01 for both) (Table 1). An older age was observed among children from the intervention group (6.8 years, IQR: 6.5 to 7.0) compared to the children from the comparison group (6.7 years, IQR: 6.4 to 6.9). Additionally, a statistical difference was observed between the groups in maternal education level with higher education among mothers from the intervention group. No differences were observed in sex, weight and height between the two groups.

Pre-intervention the intakes of the selected foods and beverages were comparable between the intervention and comparison groups (Table 1), however the intake of whole-grain bread was higher in the intervention group (group median: 13 g/d, IQR: 4 to 26) than in the comparison group (group median: 8 g/d, IQR: 3 to 17, *p* = 0.004). 

Children in the three SES groups did not differ at baseline with respect to BMI or weight (15.9 kg/m^2^ (IQR: 15.0 to 16.7), 15.7 kg/m^2^ (IQR: 14.9 to 16.6) and 15.5 kg/m^2^ (IQR: 14.7 to 16.4) (*p* = 0.19)); and the incidence of overweightness or obesity was 15.8%, 15.8% and 12.5% (*p* = 0.52) in the children from the low, middle and high SES groups, respectively (Table 2). Additionally, no differences were observed in sex, age, weight and height between the three groups.

Using linear regression analyses, no statistically significant differences in change of fruit, vegetables, sweets, SSBs, fruit juice, white bread, whole-grain bread, rye bread or fish intake were observed after 3 years of follow-up (Table 3). In adjusted models, a significant difference in change of whole-grain bread intake was observed (group means: 6.1 g/d, 95% CI: 2.2 to 10.0 in the intervention group vs. 0.3 g/d, 95% CI: −3.1 to 3.7 in comparison group) (*p* = 0.04; Table 3). No further differences were observed for intake changes (Table 3).

A significant interaction was seen between maternal education level and intervention status in relation to changes in fruit intake (*p* = 0.01). The further analyses stratified by maternal education level revealed a significant difference in changes in fruit intake between the intervention (group mean: 9.3 g/d, 95% CI: −16.1 to 94.6) and the comparison groups (group mean: −40.0 g/d, 95% CI: −56.0 to −23.9, *p* = 0.006) for those children whose mothers had short education, while no group differences were observed for those children with mothers with medium or long educations (Figure 1). No interaction effects were observed between maternal education level and intervention status in relation to changes in any of the other foods or beverages.

## 4. Discussion

This study evaluated the effect of a three-year multi-component school-based intervention on food and beverage intake and found only little indication that the overall food intake changed as a consequence of the intervention. Compared to children in the comparison group, the intake of whole-grain bread increased slightly among the children from the intervention group; no other overall differences were observed. However, the analyses also revealed that the biggest difference in fruit intake during the three-year intervention period was seen among children from low SES families, which is encouraging since children from low SES families often have a diet that is less in accordance with dietary guidelines than children from high SES families [44]. Indeed, many previous studies have found a lower consumption of healthy foods, such as fruit and vegetables, and higher consumption of unhealthy foods (e.g., SSBs) in children from low SES families [45,46]. Low SES families also tend to keep unhealthy behaviors [47]. Thus, the need for improvements in food intake generally seems greater for children from less educated families. Our findings agree with results from a previous study where improved parental knowledge concerning child nutrition among low SES families contributed to promoting the consumption of fruit [48]. Hence, the present study found that children from low SES families improved their intake of fruit and suggests that this type of multi-component school-based intervention may have reached those children most in need of dietary improvements. This SES-related change in fruit consumption during the intervention may have influenced the simultaneous differences in BMI reported earlier [40].

Insufficient fruit and vegetable consumption has been found among children with low academic achievement and at a higher risk of obesity as well as other obesity-related illnesses [45,49]. Therefore, the consumption of fruit and vegetables is critical for childhood health and development [49]. Several studies and systematic reviews have investigated school-based intervention effects on the consumption of fruit and vegetables [30,32,50,51,52,53,54,55,56]. Indeed, some previous studies found that school-based interventions increased consumption of fruit and/or vegetables in children [30,51,56], while others found no effect [32,52]. Furthermore, two studies concluded that school-based interventions seemed to improve fruit intake, but had no [53] or minimal [54] effect on vegetable intake. Our findings are in agreement with these studies. The different results may be due to different intervention content, for example, giving free fruit to increase intake of fruit [56], targeting different age groups [32] or follow-up duration [31]. Additionally, it is possible that children prefer eating fruit more than vegetables, which may make it easier to increase fruit consumption [55]. 

In the present study, we did not find significant differences in changes in the intake of SSBs, fruit juice and sweets between the intervention and the comparison group, but SSB intake decreased in both the intervention and comparison groups over the three-year intervention period. These results are in agreement with results from another study, which showed that the intake of SSBs among adolescents decreased from 2002 to 2018 [57]. In our previous study we did not show differences between the intervention and the comparison groups in relation to weight, height or skin folds, which fits the current results on dietary intake [38]. 

The present study showed a slightly higher increase in intake of whole-grain bread in the intervention group compared to the comparison group. Whole-grain bread is rich in healthy nutrients that may contribute to the prevention of several diseases, including cardiovascular disease, cancers, diabetes and gastrointestinal problems [58]. Among children, a higher consumption of whole grain was previously found to be associated with a higher intake of dietary fiber, vitamin B and minerals (Mg, Fe, P and K) [59], and was further related to lower serum insulin independent of fat mass [60]. 

It was not possible to determine which intervention initiatives were useful since the intervention was multi-component. In addition, not all initiatives were introduced on the first day of the intervention, for example, school canteens. It was assumed that school canteens and the information provided to parents about preparation of healthy food through the newsletter were the main interventions that influenced children’s dietary intake. We did not have information on the number of children using the school canteen, the number of parents receiving the newsletter or whether the canteen staff followed the recipes provided. Thus, we were unable to evaluate the effectiveness of these initiatives further. Finally, it cannot be excluded that the significant findings related to intake of whole-grain bread may be a chance finding. In any case, an increase in the intake of whole-grain bread of 6 g/day has little nutritional implication.

The strengths of the present study included that the dietary information was obtained using a 7-day food record which enabled analyses at the individual level. Moreover, we had information on maternal educational levels that allowed for stratified analyses by socioeconomic status. Additionally, since this study was initiated and handled by the municipality, our results may be applicable if a similar initiative is implemented in other schools. Furthermore, the results were adjusted for some relevant potential confounders. 

However, some limitations should also be noted: The dietary assessment used in the present study may lead to misreporting [61,62] and recall bias. Respondents may under-report their intake of unhealthy foods (e.g., sweets, SSBs) or over-report their intake of healthy foods (e.g., fruits and vegetables) [63]. A 7-day food record requires a high level of motivation and literary skill, which may affect the generalizability of the results [64]. During the recording period, people may alter the dietary behaviors to avoid a response burden, or even choose not to report their actual intake [64]. A previous study has demonstrated that parents can report food and drink intake among their preschool children with good relative accurately, while the accuracy of portion size estimates was lower [65]. Additionally, most Danish children bring packed lunch boxes from home, in contrast to many other countries, and hence child self-purchase of food at school is often limited, particularly at young ages [24]. Thus, the parents’ knowledge about their children’s lunch and snack intake during school hours may be rather good. The intervention may have increased the awareness of healthy foods among parents in the intervention group and led to parents reporting a healthier dietary intake post-intervention than was actually true, which would, if anything, have inflated difference in food intake between intervention and comparison children.

Selection bias may have affected the results due to a high dropout rate between the baseline and thethree-year follow-up. There were only a few differences in characteristics and diet intake between included and not included children. In this study, the included children consumed more vegetables and whole-grain bread, and their mothers’ BMI was lower than that of the non-included children [40]. Assuming that the more health-concerned children and parents chose to remain in the study, they therefore may have been more motivated to change their diet than those not remaining. However, as intervention effects are generally accepted by people of better affluence and education, our results are encouraging, as we found significant differences in intervention effects on the intake of fruit and vegetables confined especially to the children with less educated mothers [66]. 

## 5. Conclusions

Overall, the present study found no convincing general effect on dietary intake of introducing a multi-component school-based intervention except for a slightly increased intake of whole-grain bread in the intervention group. However, children from low SES families seemed more likely to increase their intake of fruit after the three-year multi-component school-based intervention, which seem encouraging, as children from low SES families are generally considered to be harder to reach with interventions. Future studies should aim at developing more effective ways to improve dietary intake among young school-aged children. 

## Figures and Tables

**Figure 1 ijerph-18-10543-f001:**
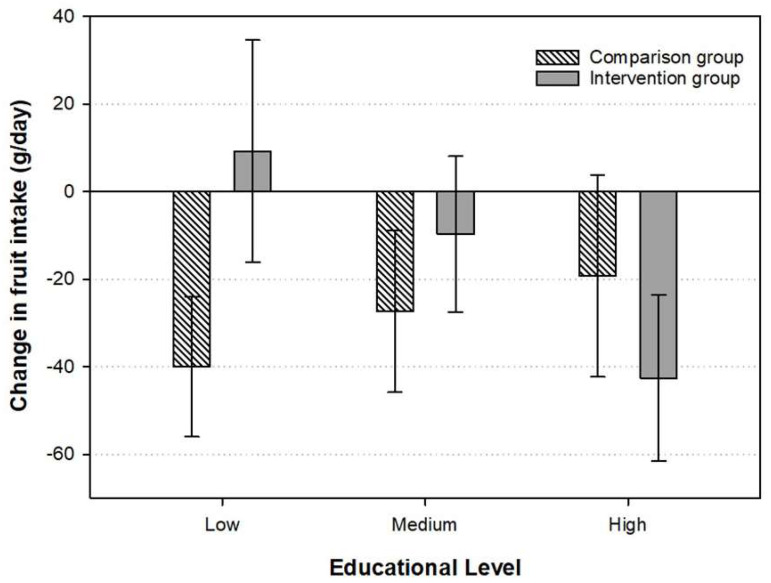
Intake of fruit in intervention/comparison groups by maternal education level. Intervention group (low education *n* = 46, medium education *n* = 64 and high education *n* = 74). Comparison group (low education *n* = 42, medium education *n* = 50 and high education *n* = 30). Adjusted for age and sex.

**Table 1 ijerph-18-10543-t001:** Pre-intervention characteristics of the participants by intervention/comparison group.

	Intervention (*n* = 184)		Comparison (*n* = 123)		*p*-Value
	Median	%/IQR	Median	%/IQR	
Boys n (%) ^a^	95	52%	52	42%	0.11
Age (y) ^b,c^	6.8	6.5; 7.0	6.7	6.4; 6.9	0.01
Weight (kg) ^b,d^	23.8	22.2; 26.3	23.7	21.6; 25.7	0.45
Height (cm) ^b,d^	123.6	120.2; 126.4	122.9	119.5; 125.4	0.15
Maternal BMI (kg/m^2^) ^b,e^	22.5	20.7; 24.7	22.3	20.8; 25.0	0.88
Maternal education, short n (%) ^a^	46	25%	43	35%	0.01
Maternal education, medium n (%)	64	35%	50	41%	
Maternal education, long n (%)	74	40%	30	24%	
Fruit (g/d) ^b^	118	68; 186	131	69; 216	0.43
Vegetables (g/d) ^b^	121	89; 167	138	88; 181	0.18
Sweets (g/d) ^b^	23	14; 35	25	16; 34	0.57
SSBs (g/d) ^b^	296	168; 409	321	171; 475	0.10
Fruit juice (g/d) ^b^	26	0; 77	26	0; 103	0.22
White bread (g/d) ^b^	49	25; 70	56	34; 76	0.24
Whole-grain bread (g/d) ^b^	13	4; 26	8	3; 17	0.004
Rye bread (g/d) ^b^	63	44; 82	62	48; 81	0.71
Fish (g/d) ^b^	9	3; 18	7	2; 16	0.32

*n*, number of observations; IQR, interquartile range; SSBs, sugar-sweetened beverages. ^a^, differences between intervention and comparison group tested by chi-squared test. ^b^, differences between intervention and comparison group tested by Wilcoxon rank-sum test; ^c^, comparison group, *n* = 122; ^d^, comparison group, *n* = 121; ^e^, intervention group, *n* = 176, and comparison group, *n* = 118.

**Table 2 ijerph-18-10543-t002:** Pre-intervention characteristics of the participants by SES.

	Short Maternal Education (*n* = 89)		Medium Maternal Education (*n* = 114)		Long Maternal Education (*n* = 104)		*p*-Value
	Median	IQR	Median	IQR	Median	IQR	
Boys n (%) ^a^	45 (50.6%)		53 (46.5%)		49 (47.1%)		0.37
Age (y) ^b^	6.8	6.5; 7.0	6.8	6.5; 7.0	6.7	6.5; 6.9	0.54
Weight (kg) ^b^	23.4	22.2; 26.4	24.2	22.2; 26.3	23.2	21.5; 25.6	0.09
Height (cm) ^b^	122.9	119.7; 125.5	123.8	120.2; 127.6	123.0	119.7; 125.4	0.21
BMI (kg/m^2^) ^b^	15.9	15.0; 16.7	15.7	14.9; 16.6	15.5	14.7; 16.4	0.19
Overweight and obesity n (%) ^b^	14 (15.8%)		18 (15.8%)		13 (12.5%)		0.52
Physical activity (cpm) ^b^	736.3	606.5; 885.1	709.5	596.5; 837.8	674.0	590.6; 829.8	0.24

^a^, differences between intervention and comparison group tested by chi-squared test. ^b^, differences between intervention and comparison group tested by Wilcoxon rank-sum test.

**Table 3 ijerph-18-10543-t003:** Effects (changes between pre- and post-intervention) of the intervention on intake of foods and beverages (g/d) in the intervention and in the comparison group ^a^.

	Model	*n*	Intervention Group		Comparison Group		*p*-Value
			Mean	95% CI	Mean	95% CI	
Fruit (g/d)	Crude ^b^	307	−17.7	−28.7; 6.6	−30.9	−41.9; −19.9	0.08
	Adjusted ^c^	306 ^d^	−17.1	−30.2; −4.1	−31.3	−42.3; −20.2	0.12
Vegetables (g/d)	Crude	307	−13.5	−21.1; −5.9	−21.5	−32.5; −10.5	0.17
	Adjusted	306	−15.1	−23.0; −7.2	−19.2	−29.8; −8.7	0.57
Sweets (g/d)	Crude	307	0.9	−1.2; 3.0	1.4	−2.5; 5.3	0.85
	Adjusted	306	0.7	−1.6; 3.1	1.7	−1.8; 5.2	0.65
SSBs (g/d)	Crude	307	−106.2	−128.9; −83.5	−89.3	−129.3; −49.2	0.52
	Adjusted	306	−105.8	−135.1; −76.4	−88.5	−120.7; −56.3	0.46
Fruit juice (g/d)	Crude	307	18.1	4.9; 31.4	31.6	11.7; 51.6	0.29
	Adjusted	306	18.2	4.4; 32.1	31.3	12.5; 50.1	0.27
White bread (g/d)	Crude	307	12.1	7.1; 17.1	9.2	4.2; 14.2	0.42
	Adjusted	306	11.3	5.6; 17.0	10.4	4.0; 16.7	0.84
Whole-grain bread (g/d)	Crude	307	5.7	1.8; 9.7	0.9	−2.6; 4.3	0.07
	Adjusted	306	6.1	2.2; 10.0	0.3	−3.1; 3.7	0.04
Rye bread (g/d)	Crude	307	−2.3	−6.8; 2.2	−0.2	−4.8; 4.4	0.48
	Adjusted	306	−2.5	−7.0; 2.0	0.02	−4.2; 4.3	0.46
Fish (g/d)	Crude	307	−0.2	−1.9; 1.5	−1.8	−3.6; −0.5	0.16
	Adjusted	306	−0.2	−2.0; 1.5	−1.6	−3.0; −0.2	0.27

^a^, difference between groups was tested using linear regression modelling. ^b^, crude models were conducted with information on outcome, intervention status and baseline measure of outcome, only. ^c^, adjusted analyses with added information on sex, age and maternal education were carried out. ^d^, one child did not have information on age.

## Data Availability

The data presented in this study are available on request to bfh-dl-eek@regionh.dk. The data are not publicly available due to the participants’ privacy and data protection.

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
