# Peer review of "Three-Year Intervention Effects on Food and Beverage Intake—Results from the Quasi-Experimental Copenhagen School Child Intervention Study (CoSCIS)"

_ijerph, 2021, doi:10.3390/ijerph181910543_

Round 1

Reviewer 1 Report

The study by Ren et al. on Danish the nutrition of school children is interesting in terms of how good are interventions for potential change of eating behavior in a given cohort of children.

However, some considerations about the current version of the manuscript have to be made:

  1. Apart from the change in nutrition in any nutrition intervention study in children one is concerned with the question: did the intervention change anything during the time studied in their growth pattern, particularly if one is concerned about the enormous increase of obesity in young children. In the method section of the paper it is stated that anthropometric data were evaluated in all children pre- and post- intervention. In the table 1 only the pre-intervention data of height and weight are described. It would be interesting to see the change of z-scores of weight and height before and after intervention, whether there was a difference in the three groups.  Even if these questions have been addressed in previous studies, they should be at least mentioned and referred to existing data.
  2. Also missing is a description of the three groups prior to start according to the presence of overweight and obesity. What was the rate of overweight and obesity in these children before and after the intervention?
  3. If there was overweight and obesity in some children, what was influence of parent education? Was the number of overweight and obese children higher in the low SES group?
  4. Daily total fluid intake is also very important in children. Has the daily total fluid intake been assessed? And if so what were the differences? What was the total fluid intake of noncaloric drinks like water? What about milk?
  5. What about other soft drinks except fruit juices?

Author Response

  1. Apart from the change in nutrition in any nutrition intervention study in children one is concerned with the question: did the intervention change anything during the time studied in their growth pattern, particularly if one is concerned about the enormous increase of obesity in young children. In the method section of the paper it is stated that anthropometric data were evaluated in all children pre- and post- intervention. In the table 1 only the pre-intervention data of height and weight are described. It would be interesting to see the change of z-scores of weight and height before and after intervention, whether there was a difference in the three groups.  Even if these questions have been addressed in previous studies, they should be at least mentioned and referred to existing data.

The effect of the intervention on anthropometry has been explored previously [38], we now mention these results in the manuscript (line 305)  “In our previous study we did not show differences between the intervention and the control groups in relation to weight, height or skin folds, which fits the current results on dietary intake [38].”

  1. Also missing is a description of the three groups prior to start according to the presence of overweight and obesity. What was the rate of overweight and obesity in these children before and after the intervention?

Thanks for this comment. Children did not differ in BMI, weight, age or sex, and we have now added a Table 3 and a paragraph in the results section about this: “Children in the 3 SES groups did not differ at baseline with respect to BMI or weight (15.9 kg/m2 (IQR 15.0 to 16.7), 15.7 kg/m2 (IQR 14.9 to 16.6) and 15.5 kg/m2 (IQR 14.7 to 16.4) (p= 0.19); and overweight or obese was 15.8%, 15.8% and 12.5% (p=0.52) in the children from the low, middle and high SES groups, respectively (Table 2). Also, no differences were observed in sex, age, weight and height between three groups.” (line 205)

  1. If there was overweight and obesity in some children, what was influence of parent education? Was the number of overweight and obese children higher in the low SES group?

Thanks for this comment. Children did not differ in the number of obesity and we have now added a Table 3 and a paragraph in the results section about this: “Children in the 3 SES groups did not differ at baseline with respect to BMI or weight (15.9 kg/m2 (IQR 15.0 to 16.7), 15.7 kg/m2 (IQR 14.9 to 16.6) and 15.5 kg/m2 (IQR 14.7 to 16.4) (p= 0.19); and overweight or obese was 15.8%, 15.8% and 12.5% (p=0.52) in the children from the low, middle and high SES groups, respectively (Table 2). Also, no differences were observed in sex, age, weight and height between three groups.” (line 205)

  1. Daily total fluid intake is also very important in children. Has the daily total fluid intake been assessed? And if so what were the differences? What was the total fluid intake of noncaloric drinks like water? What about milk?

Unfortunately total fluid intake was not assessed, mainly because water intake is poorly assessed. However, we acknowledge that our choice of outcome measures is not sufficiently justified, and we have now added the following “The food and beverages groups included in the present study was based on the Danish food-based dietary guidelines [42,43].” (line 158)

  1. What about other soft drinks except fruit juices?

we acknowledge that our choice of outcome measures is not sufficiently justified, and we have now added the following “The food and beverages groups included in the present study was based on the Danish food-based dietary guidelines [42,43].” (line 158)

Reviewer 2 Report

In the article "Three-Year Intervention in Food and Drink Consumption - Results of the Quasi-experimental Copenhagen School Child Intervention Study (CoSCIS)," the authors assessed the effect of a three-year multi-component school intervention on food and drink consumption. The assumption of the research is very important due to the growing problem of obesity among children and adolescents. However, the time that elapsed from the time the study was conducted to its publication is disturbing, as it amounted to 16 years.
In my opinion, the work is suitable for publication after corrections have been made, especially in such matters as:
-in Materials and Methods: too many references to previous publications make it difficult to verify both the research group and the research methodology, expand this part.
-Table 1 - illegible due to indexes marked with symbols (†, ‡, §, |), Please enter letters or numbers
-Figure 2 - illegible, please correct it.
- some technical errors, e.g. in lines 118 and 216 different case of letters in the word Figure, please correct.

Author Response

  1. In Materials and Methods: too many references to previous publications make it difficult to verify both the research group and the research methodology, expand this part.

Thanks for your comments, we have expanded the methods section (line 104, line 138, line 158, line 163).

Lines 138-42:

“The food records were divided into 4 parts: breakfast, lunch, dinner and snacks. Each meal was further divided into beverages, bread, cereals, vegetables etc. Household measurements and photo series illustrating the serving sizes and 4 or 6 different quantities of common foods were used to assess the amount of food consumed”

Lines 158- 59

“The food and beverages groups included in the present study were based on the Danish food-based dietary guidelines [42,43].”

Line 163-65

“The first question was “what school education have you achieved or are you currently completing?”. The second question was “do you have a vocational education?”.

  1. -Table 1 - illegible due to indexes marked with symbols (†, ‡, §, |), Please enter letters or numbers

Thank you, we have changed those to letters (line 213, line 240)

  1. -Figure 2 - illegible, please correct it.

We are happy to correct any issues with the figure, but we are not sure what the issue is. We have now made a new figure, which complies to all format request mentioned on https://www.mdpi.com/journal/ijerph/instructions#figures .  If the figure still doesn’t fulfill  the journal requirements, please elaborate.

  1. - some technical errors, e.g. in lines 118 and 216 different case of letters in the word Figure, please correct.

Thanks for your comments, we have corrected this error, in line 126 (Figure S1) and line 256 (Figure 1).

Reviewer 3 Report

The manuscript by Driessen-Ren, X. et al. entitled “Three-year intervention effects on food and beverage intake - results from the Quasi-experimental Copenhagen School Child Intervention Study (CoSCIS)” is part of the results from the bigger study (CoSCIS intervention) which has the aim to improve diet and increase physical activity among the local school children in Denmark. The goal was to investigate impact of intervention (education) on children eating habits and whether the mother 's education will have an impact on different (higher) intake of fruit and vegetables and a lower intake of sweets and soft drinks than the children compared to control

The manuscript is understandably and systematically written. The problem for the reader is that a lot of data is published elsewhere thus losing reading fluency of this manuscript. Topic is very important nowadays because of the large increase in the number of the overweight and obese children.

Some technical comments:

Row 52 – A healthy diet, with regular physical activity

Row 56 - consumption of meat (emphasize that it is red meat, not meat per se)

Row 103 - which anthropometric measures and which instruments were used, also there is no data how did you measure or assessed physical activity

Can you give explanation of 2 categories of short education under the Socioeconomic status section

Row 184 – table one - maybe it will not bad to distribute the most important results by gender because it turns out that in the intervention group there are mostly boys and that they have average BM almost the same as the one (comparison group) where are equally girls and boys. Question for the same table is why there are not data such as dietary fibre intake, as well as the intake of some crucial vitamins and minerals important for children population such as Ca, Fe, D, A, C

Row 282/283 – you stated that /DD is your strength of the study but that can also be great weakness because people have tendency of change their diet during the 7DD, they usually alters his or her eating patterns or food choices during the period of recording so that it does not reflect usual intake.

English proofreading is suggested.

Author Response

The manuscript is understandably and systematically written. The problem for the reader is that a lot of data is published elsewhere thus losing reading fluency of this manuscript. Topic is very important nowadays because of the large increase in the number of the overweight and obese children.

 Thanks for this comment, which was also made by reviewer 2. Accordingly, we have now expanded the methods section (line 104, line 138, line 158, line 163).

Lines 138-42:

“The food records were divided into 4 parts: breakfast, lunch, dinner and snacks. Each meal was further divided into beverages, bread, cereals, vegetables etc. Household measurements and photo series illustrating the serving sizes and 4 or 6 different quantities of common foods were used to assess the amount of food consumed”

Lines 158- 59

“The food and beverages groups included in the present study were based on the Danish food-based dietary guidelines [42,43].”

Line 163-65

“The first question was “what school education have you achieved or are you currently completing?”. The second question was “do you have a vocational education?”.

Some technical comments:

  1. Row 52 – A healthy diet, with regular physical activity

We have added this sentence (line 51)

  1. Row 56 - consumption of meat (emphasize that it is red meat, not meat per se)

We have emphasized “red meat” at line 56

  1. Row 103 - which anthropometric measures and which instruments were used, also there is no data how did you measure or assessed physical activity

Thanks for your comments, we have described the measurements and expanded the methods section.

Lines 104-12 “Weight and height were measured at enrollment by trained researchers. Height was measured to the nearest 1 mm by a Harpenden stadiometer. Weight was measured to the nearest 0.1 kg by an electronic scale (Seca 882, Medical Scales, NY, USA). Habitual physical activity was measured in 4 consecutive days (2 weekendays+ 2 weekdays) by the MTI 7164 activity monitor (Manufactory Technology Inc., Fort Walton Beach, FL, USA). Children were required to wear accelerometers at all times, except for sleep and water activities. Only children with a cumulative activity time of more than 8 hours per day for at least 3 days were included in the analysis.”

Lines 138-42:

“The food records were divided into 4 parts: breakfast, lunch, dinner and snacks. Each meal was further divided into beverages, bread, cereals, vegetables etc. Household measurements and photo series illustrating the serving sizes and 4 or 6 different quantities of common foods were used to assess the amount of food consumed”

Lines 158- 59

“The food and beverages groups included in the present study is based on the food-based dietary guidelines [42,43].”

Line 163-65

“The first question was “what school education have you achieved or are you currently completing?”. The second question was “do you have a vocational education?”.

  1. Can you give explanation of 2 categories of short education under the Socioeconomic statussection

The second short education is vocational training e.g. health care worker, laboratory technician or dental hygienist, in the manuscript. We have changed the second “short education” to “vocational education” (now specified in line 169)

  1. Row 184 – table one - maybe it will not bad to distribute the most important results by gender because it turns out that in the intervention group there are mostly boys and that they have average BM almost the same as the one (comparison group) where are equally girls and boys. Question for the same table is why there are not data such as dietary fibre intake, as well as the intake of some crucial vitamins and minerals important for children population such as Ca, Fe, D, A, C

As indicated by the p-value there was no statistically significant difference in gender distribution across groups, and we furthermore see no obvious reason why the intervention should have a different effect on boys than girls, Thus, we have chosen not to stratify the analyses. With regards to additional analyses, although interesting, all outcomes were predefined. In line with good scientific practice, we would prefer not to deviate too much from our original plan, unless there is an extremely strong rationale for doing so. The focus of this paper is food groups selected based on the dietary recommendations, and thus exploring specific micronutrient is beyond the purpose and scope of our study.

  1. Row 282/283 – you stated that /DD is your strength of the study but that can also be great weakness because people have tendency of change their diet during the 7DD, they usually alters his or her eating patterns or food choices during the period of recording so that it does not reflect usual intake.

Thanks for your comments, we have now clarified and discussed in more detail about dietary record in line 335. “Some respondents may under-report intake of unhealthy foods (e.g. sweets, SSB) or over-report intake of healthy foods (e.g. fruits and vegetables).” “During the recording period, people may alter the dietary behaviors to avoid a response burden, or even choose not to report actual intake. This would lead to inaccuracies in recording that most likely would have attenuated the observed results”

  1. English proofreading is suggested.

Thanks for your comments, we have checked and corrected our manuscript.